# Pathologic Proteolytic Processing of N-Cadherin as a Marker of Human Fibrotic Disease

**DOI:** 10.3390/cells11010156

**Published:** 2022-01-04

**Authors:** Paul Durham Ferrell, Kristianne Michelle Oristian, Everett Cockrell, Salvatore Vincent Pizzo

**Affiliations:** 1Department of Pathology, Duke University Medical Center, Durham, NC 27710, USA; paul.ferrell@duke.edu (P.D.F.); rettcockrell@gmail.com (E.C.); 2Department of Radiation Oncology, Duke University Medical Center, Durham, NC 27710, USA; kristi.oristian@duke.edu

**Keywords:** fibrosis, heart failure, pro-N-cadherin, fibronectin, myofibroblasts, N-cadherin, cirrhosis, pulmonary fibrosis

## Abstract

Prior research has implicated the involvement of cell adhesion molecule N-cadherin in tissue fibrosis and remodeling. We hypothesize that anomalies in N-cadherin protein processing are involved in pathological fibrosis. Diseased tissues associated with fibrosis of the heart, lung, and liver were probed for the precursor form of N-cadherin, pro-N-cadherin (PNC), by immunohistochemistry and compared to healthy tissues. Myofibroblast cell lines were analyzed for cell surface pro-N-cadherin by flow cytometry and immunofluorescent microscopy. Soluble PNC products were immunoprecipitated from patient plasmas and an enzyme-linked immunoassay was developed for quantification. All fibrotic tissues examined show aberrant PNC localization. Cell surface PNC is expressed in myofibroblast cell lines isolated from cardiomyopathy and idiopathic pulmonary fibrosis but not on myofibroblasts isolated from healthy tissues. PNC is elevated in the plasma of patients with cardiomyopathy (*p* ≤ 0.0001), idiopathic pulmonary fibrosis (*p* ≤ 0.05), and nonalcoholic fatty liver disease with cirrhosis (*p* ≤ 0.05). Finally, we have humanized a murine antibody and demonstrate that it significantly inhibits migration of PNC expressing myofibroblasts. Collectively, the aberrant localization of PNC is observed in all fibrotic tissues examined in our study and our data suggest a role for cell surface PNC in the pathogenesis of fibrosis.

## 1. Introduction

Fibrosis is a major contributor to the chronic decline of organ function associated with end-stage organ failure. It is not unique to a single organ but manifests in any organ in the human body [1]. In the United States, forty-five percent of all deaths can be attributed to fibrosis-related disease [2,3,4]. Fibrosis is described as the remodeling of tissue architecture which results in loss of organ function due to the substitution of the functional parenchyma with mesenchymal tissue. While transient tissue fibrosis is a normal response to injury during wound healing, pathological fibrosis is characterized by relentless, non-resolving extracellular matrix (ECM) deposition and progressive tissue remodeling. Fibrosis progresses when a poorly characterized wound healing process is activated, leading to excessive remodeling and deposition of ECM by myofibroblasts, primarily fibronectin, type I and type III collagen [5,6]. Myofibroblasts are both ubiquitous and invasive as part of canonical wound healing; however, the origin of the myofibroblasts that fail to undergo programmed cell death or dedifferentiation following wound healing and result in progressive fibrosis remains ambiguous [7,8].

To date, there is no effective therapeutic to halt or reverse fibrotic progression and tractable therapeutic targets with pathological specificity are extremely limited [5,9,10]. Only two therapeutics, Nintedanib and Pirfenidone, have been approved for fibrosis, and in only one organ system, the lungs. However, even with treatment idiopathic pulmonary fibrosis (IPF) continues to progress [4]. Understandably, there have been several attempts at therapeutic intervention designed to target proteins differentially expressed by myofibroblasts, including several successful attempts at abolishing fibrosis in rodent models [9,11]. Notably, translating these models to the clinic has been challenging [4]. The salient example of this is the significant number of compounds targeting TGF-β1 and its associated signaling. TGF-β1 is a critical growth factor with roles in normal wound healing and immunity, as well as cancer progression, myofibroblast activity and fibrogenesis [9]. Neutralizing TGF-β1 signaling in rodent models abolishes fibrosis; however, molecules targeting TGF-β1 signaling in humans have failed to progress clinically [11]. These failures to translate preclinical findings to the clinic represent an unmet need in the field to better identify therapeutic targets with specificity to fibrotic tissues.

Here we present data demonstrating that retention of the N-cadherin prodomain at the cell surface is a potential biomarker of pathological myofibroblasts and fibrosis associated tissues of the heart, lungs, and liver. Canonical N-cadherin protein processing occurs in the transgolgi network after the precursor N-cadherin is translated with an N-terminal precursor prodomain, termed pro-N-cadherin (PNC) [10,12,13]. The precursor prodomain is then cleaved by proprotein convertases, generating the active mature N-cadherin protein which functions at the cell surface by forming homodimers at cellular adherens junctions [10,12]. The prodomain is non-adhesive and sterically restricts N-cadherin dimer formation by inhibiting N-cadherin tryptophan swapping [12]. During certain developmental and pathological processes, PNC is localized to the cell surface with mature N-cadherin [14,15,16]. To date, cell surface PNC expression has been observed in only two contexts: Perinatal synapse formation in vertebrates, and carcinogenesis [14,15,16]. In this work, we identify a third clinically significant context in which PNC is expressed on the surface of myofibroblasts from pathological origins and aberrantly localized on tissues from failing heart, lung, and liver.

## 2. Materials and Methods

### 2.1. Cell Culture and Reagents

All cells were cultured at 37 °C, 5% CO_2_ in a humidified chamber. Cells were used between passages 3 and 10 and validated for alpha-smooth muscle actin (α-SMA) expression prior to use in experiments by immunoblot using α-SMA antibody (Cell Signaling 19245S, Danvers, MA, USA). The LL97A (ATCC CCL-191, Manassas, VA, USA), LL29 (ATCC CCL-134, Manassas, VA, USA) and CCD-16Lu (ATCC CCL-204, Manassas, VA, USA) cell lines were purchased from ATCC and grown by their specifications. Primary ventricular normal human cardiac fibroblasts (NHCF) from a healthy donor heart were purchased from Lonza (CC-2904, Morrisville, NC, USA) and grown in their recommended media. Primary ventricular cardiac fibroblasts (DCM-CF) from explant failed human hearts were isolated from tissue obtained from the Duke Human Heart Repository (IRB # Pro87831) and grown in DMEM;10% FBS; 1x Pen Strep. Primary normal human lung fibroblasts (NHLF) were purchased from the Duke Cell Culture Facility (Lonza CC-2512) and grown in MEM; 10% FBS; 1x Pen Strep.

### 2.2. pro-N-Cadherin Antibody Purification

pro-N-cadherin monoclonal antibody-producing hybridoma clone 10A10 was a gift from the Wahl Laboratory at the University of Nebraska. Cells were acclimated to Hybridoma-SFM (Gibco, Grand Island, NE, USA) supplemented with 10% super low IgG HI-FBS (Hyclone, Logan, UT, USA) and grown to confluency. Hybridoma cells were passaged, and the media was replaced with serum free Hybridoma-SFM media and allowed to express for 5 days. Supernatant was collected by centrifugation, passed through a 0.22 µm filter and antibodies were purified by affinity chromatography using protein G sepharose prepacked column (GE Healthcare, Salt Lake City, UT, USA) following the manufacturer’s protocol using the Amersham Biosciences AKTA FPLC system. Antibodies were dialyzed into final buffer PBS pH 7.4 and stored at −20 °C for these studies.

### 2.3. Fibroblast Isolation from Explant Tissues

Tissues were minced into small pieces and enzymatically digested in appropriate volume of tissue dissociation solution comprised of PBS pH 7.4, 5 mg/mL Collagenase Type IV (Gibco, Grand Island, NE, USA), 1.3 mg/mL Dispase II (Gibco, Grand Island, NE, USA), 0.05% Trypsin, with agitation at 37° Celsius for 1 h. After incubation in dissociation solution, 25 mL of HBSS (Gibco, Grand Island, NE, USA) was added, followed by serial pipetting for manual dissociation and centrifugation at 1200 RPM, 4 °C, for 10 min. The supernatant was then aspirated and the pellet was resuspended in 5 mL HBSS then passed through a 100 µm filter followed by a 70 µm filter. The filtered cell suspension was then centrifuged at 1200 RPM, 4 °C, for 5 min, the supernatant was aspirated and the pellet resuspended into DMEM; 10% FBS; 1x Pen Strep. Cells were plated and passaged to remove plasma and myocyte contaminant.

### 2.4. Tissue Procurement and Processing

Fresh explant cardiac tissues were received from the Duke HHR (IRB Pro00087831) within 12 h of explant in PBS pH 7.4, 1x Pen Strep (Gibco, Grand Island, NE, USA), on ice and processed same day. Tissues were divided and a portion was immediately processed for fibroblast isolation (above). The remaining portion was fixed overnight at room temperature in 4% paraformaldehyde and PBS pH 7.4 followed by 70% EtOH and embedded in paraffin. Additional flash-frozen cardiac tissues from the Duke HHR, including a subset of previously characterized samples [17] were embedded in OCT medium and sectioned on a Leica cryostat into 5 µm sections and adhered to charged glass slides for immunohistochemical analysis. Liver tissue specimens were provided by the DUHS Nonalcoholic Fatty Liver Disease Research Database and Specimen Repository under Pro00005368 and Abdominal Transplant Repository under Pro00107246. Formalin-fixed paraffin embedded heart and liver tissues were cut on a Leica microtome into 3–5 µm sections and adhered to charged glass slides for immunohistochemical analysis. Lung tissue sections were obtained from the Duke Bio Repository and Precision Pathology Center (BRPC) on charged glass slides under an IRB exemption. Additional heart, lung, and liver tissue microarrays were obtained commercially from USA Biomax, Inc. (Rockville, MD, USA) as formalin-fixed paraffin embedded 1.5 mm cores on charged glass slides.

### 2.5. Immunohistochemistry

Immunohistochemical detection of PNC was performed on formalin fixed paraffin embedded tissue samples sectioned at 3–5 µm. Sections were deparaffinated with xylene, rehydrated, and treated with 3% H_2_O_2_ to quench endogenous peroxidase. Heat-mediated antigen retrieval was performed in a citrate buffer (pH 6) and blocked with 5% horse serum. PNC was detected using purified m-α-PNC mAb clone 10A10 at 5–10 µg/mL at 4 °C overnight. An avidin-biotin amplification step and chromogenic detection (DAB) of α-mouse HRP-conjugated secondary antibody was used to visualize pro-N-cadherin localization and expression. Tissues were counter-stained with Mayer’s hematoxylin and mounted with Cytoseal 60 (Thermo Fisher, Grand Island, NE, USA) mounting media for imaging.

### 2.6. Plasma Procurement

Healthy human donor plasma was purchased commercially from Innovative Research, Inc, Novi, MI, USA. Plasma from heart failure patients was provided by the Duke HHR IRB Pro00087831. Plasma from NAFLD-cirrhosis patients was provided by the Duke Nonalcoholic Fatty Liver Disease Research Database and Specimen Repository under Pro00005368 and Abdominal Transplant Repository under Pro00107246. Plasma from IPF patients was purchased commercially from Innovative Research, Inc.

### 2.7. Immunoprecipitations

LL29 conditioned media was incubated at 4 °C with 10 µg of murine 10A10 antibody for 1 h. Protein G Sepharose (GE Healthcare, Salt Lake City, UT, USA) was blocked with 1% BSA in PBS then 5 µL of resin was added to the conditioned media. After incubating at 4 °C for 30 min, the resin was washed three times with PBS then eluted using 1x LDS sample buffer (Thermo Fisher, Grand Island, NE, USA) with B-mercaptoethanol. Elutes were boiled, immunoblotted, and developed using humanized 10A10 chimeric antibody.

For plasma samples, 10A10 and mouse IgG1 isotype control (Thermo Fisher, Grand Island, NE, USA) antibodies were first crosslinked to protein G sepharose (GE Healthcare, Salt Lake City, UT, USA) using Pierce crosslink IP kit (Thermo Fisher, Grand Island, NE, USA). Equivalent volumes of healthy donor and diseased donor plasma samples were pooled then diluted 1:10 in PBS, centrifuged at 16,000 G and filtered using a 0.22 µm filter. Samples were then rotated overnight with antibody crosslinked protein G sepharose, washed and eluted following the manufacturer’s protocol. Elutes were immunoblotted and developed using murine 10A10 antibody.

### 2.8. SDS-PAGE and Immunoblotting

Total cell lysates were prepared using RIPA buffer adjusted to 1% *w/v* SDS (Sigma, Saint Louis, USA) supplemented with Halt Protease & Phosphatase Inhibitor Cocktail (Thermo Fisher, Grand Island, NE, USA) and Benzonase (Millipore Sigma, Saint Louis, MO, USA). Cell surface proteins were isolated using the Pierce cell surface biotinylation and isolation kit (Thermo Fisher, Grand Island, NE, USA) following the manufacturer’s protocol. Plasma membrane loading control Na,k-ATPase α-1 (cell signaling 23565T, Massachusetts, USA) was used for cell surface protein isolates. Protein concentration of each lysate was measured using Pierce BCA protein assay (Thermo Fisher, Grand Island, NE, USA). For total lysates, precleared lysates were boiled in sample buffer (Thermo Fisher, Grand Island, NE, USA) and 40 µg of protein was loaded. All samples were run on 10% NuPage gels containing 0.1% SDS under reducing conditions. A discontinuous Laemmli buffer system was used. The proteins were transferred from the gels to nitrocellulose membranes. The molecular weights were assessed using Precision Plus Prestained Marker (Bio-Rad, Hercules, CA, USA). The membranes were thoroughly washed with tris-buffered saline (TBS) and then blocked with infrared blocking buffer (Rockland, Pottstown, PA, USA) for 1 h at room temperature. Membranes were incubated with antibodies overnight at 4 °C in 5% BSA, 1x TBS, 0.1% Tween-20. RPL13A (Cell Signaling 2765S, Danvers, MA USA) was used for total cell lysate loading control, along with mature N-cadherin (Sigma, GC-4 clone, C3865, Saint Louis, MO, USA) and α-PNC mAb. After incubation with the primary antibody, the membranes were washed three times for 5 min each with 1x TBS containing 0.1% Tween 20 (TBST). The membranes were then incubated with the manufacturer’s recommended dilution of appropriate Alexa Fluor conjugated secondary (Thermo Fisher, Grand Island, NE, USA). The membranes were washed twice for 5 min each with TBST and once with TBS for 5 min. The probed membranes were scanned on a Li-Cor Odyssey System (Li-Cor Biosciences, Lincoln, NE, USA).

### 2.9. Flow Cytometry

Cells were plated in complete media in 10 cm dishes and allowed to anchor overnight. The following day, cells were washed with PBS pH 7.4, then detached using PBS pH 7.4; 0.5 mM EDTA; 10% Glycerol at 37 °C for approximately 5–10 min or until cell rounding followed by scraping. Cells were kept on ice for the duration of the staining procedure. Cells were pelleted at 1200 RPM, 4 °C, for 5 min followed by supernatant aspiration, PBS pH 7.4 wash, and resuspension in PBS pH 7.4; 1% BSA; 0.09% sodium azide. Cells were incubated with either α-N-cadherin antibody (Sigma, GC-4 clone, C3865), α-PNC antibody (5 µg/mL), or 5 µg/mL mouse IgG1 isotype control (Thermo Fisher, Grand Island, NE, USA) for 30 min. Cells were washed with PBS pH 7.4; 1% BSA; 0.09% sodium azide followed by 5 µg/mL Alexa Fluor 488 secondary antibody (Thermo Fisher, Grand Island, NE, USA) incubation for 30 min. Cells were then washed with PBS pH 7.4; 1% BSA; 0.9% sodium azide and stained with 7AAD (BioLegend, San Diego, CA, USA) following the manufacturer’s protocol. Cells were analyzed using the Guava EasyCyte (Luminex, Austin, TX, USA) flow cytometer and the latest version of Flowjo software, gating and excluding 7AAD positive cells. At least 20,000 events were collected for each experiment. To calculate background, the same sample was ran using the isotype control antibody four times independently and histograms were stacked to determine baseline chi-square and SE dymax % positive.

### 2.10. Immunofluorescent Microscopy

For immunofluorescent microscopy, cells were plated into multi-well chamber slides coated with 5 µg/mL human Collagen type I (Sigma, Saint Louis, MO, USA) in PBS. After cells were allowed to adhere overnight, each well was aspirated, washed with PBS pH 7.4 and fixed in 1% formaldehyde for 30 min at room temperature. For perinuclear PNC staining, cells were permeabilized using 0.05% Triton in PBS. Cells were then blocked with 5% goat serum (Abcam, Cambridge, UK); PBS pH 7.4 for 1 h at room temperature. Primary antibody against pro-N-cadherin was incubated at 2 µg/mL on cells overnight at 4 °C followed by Qdot 655 conjugated secondary (Thermo Fisher, Grand Island, NE, USA) for 1 h at room temperature. For colocalization, FN1 antibody (Cell Signaling 26836S, Danvers, MA, USA) was used following the manufacturer’s protocol followed by AlexaFluor 488 conjugated secondary (Thermo Fisher, Grand Island, NE, USA). DAPI was used following the manufacturers protocol (Sigma, Saint Louis, MO, USA) to visualize DNA. Coverslips were applied using 50% glycerol in PBS and sealed with nail polish. Images were taken using the Leica DMI400 B.

### 2.11. Sandwich Enzyme-Linked Immunosorbent Assay

Recombinant prodomain (rPro) of N-cadherin, amino acids 26–159 (Accession # AAB22854) was generated and supplied by GenScript (Piscataway, NJ, USA) and used to optimize a PNC sandwich enzyme-linked immunosorbent assay (ELISA) and later used as an antagonist to cell surface PNC in migration assays. High binding ELISA plates (Costar, Kennebunk, ME, USA) were used to bind 1 µg/well of α-PNC antibody 10A10 as the capture antibody. Washes were performed using PBS pH 7.4 0.1% Tween 20. Three washes (300 µL/well) were performed between each of the following steps using Biotek (Winooski, VT, USA) ELx405 Select CW automated plate washer. All steps were performed at room temperature with room temperature equilibrated buffers. Capture antibody was bound overnight at room temperature in PBS pH 7.4 followed by blocking with 300 µL per well blocking buffer 5% non-fat dry milk (Bio-Rad, Hercules, CA, USA) in 1x PBS (Gibco, Grand Island, NE, USA) with 0.1% Tween 20 for 1 h. Plasma samples and rPro analyte standard were applied 100 µL per well in 1% BSA, PBS pH 7.4, 5 mM EDTA, 0.1% Tween 20 for 1 h followed by 100 µL/well 1:800 dilution of biotinylated polyclonal sheep α-PNC detection antibody (R&D BAF1388, Minneapolis, MN, USA) in 1% BSA, PBS pH 7.4, 0.1% Tween 20 for 1 h. Streptavidin horseradish peroxidase conjugate (Thermo Fisher, Grand Island, NE, USA ) was applied at 100 µL per well, 1:800 in 2% BSA, PBS pH 7.4 for 20 min followed by 150 µL of ABTS (Thermo Fisher, Grand Island, NE, USA ) for 15 min and read using the Biotek (Winooski, VT, USA) Cytation 3 Imager Reader at absorbance (Abs) 410 nm.

### 2.12. Solid Phase Enzyme Immunoassays

To find rPro binding partners, a solid phase enzyme immunoassay was used. Medium binding ELISA plates (Costar, Kennebunk, ME, USA) were coated with either 10 µg/mL human plasma fibronectin (Sigma, Saint Louis, MO, USA), human type I collagen (Sigma, Saint Louis, MO, USA) or human type III collagen (Sigma, Saint Louis, MO, USA) overnight at room temperature in PBS. All washes were done with PBS 0.1% Tween-20, 300 µL/well, three times using the Biotek ELx405 Select CW automated plate washer. After immobilizing fibronectin, collagen type I, or BSA, wells were washed then blocked using 300 µL/well 1% BSA PBS in wash buffer for 1 h at room temperature. Wells were washed then a serial dilution starting at 10 µg/mL of His tagged rPro peptide diluted in blocking buffer was incubated over the immobilized proteins for one hour at room temperature. Wells were washed and incubated with 1:2000 mouse α-His biotinylated antibody (Invitrogen MA1-21315-BTIN, Grand Island, NE, USA) in block for 1 h at room temperature followed by 1:2000 streptavidin HRP (Thermo Fisher, Grand Island, NE, USA) in 2% BSA PBS at room temperature for 15 min. Immunoassay was developed using TMB Ultra (R&D, Minneapolis, MN, USA) following the manufacturers recommended protocol. Absorbance 450 nm was read using the Biotek Cytation 3 indicating bound his tagged rPro peptide.

For antibody displacement assays, medium binding ELISA plates (Costar, Kennebunk, ME, USA) were coated with human plasma fibronectin (Sigma, Saint Louis, MO, USA) at 10 µg/mL in PBS overnight at room temperature. All washes were done with PBS 0.1% Tween-20, 300 µL/well, three times using the Biotek ELx405 Select CW automated plate washer. After immobilizing fibronectin, wells were washed then blocked using 300 µL/well 5% human plasma in wash buffer for 1 h at room temperature. While blocking, antibodies were added at the respective concentration to blocking buffer containing 1.5 µg/mL recombinant his-tagged prodomain. Wells were washed and samples were applied for 1 h at room temperature. Wells were washed and incubated with 1:2000 mouse α-His biotinylated antibody (Invitrogen MA1-21315-BTIN, Grand Island, NE, USA) for 1 h at room temperature followed by 1:2000 streptavidin HRP (Thermo Fisher, Grand Island, NE, USA) at room temperature for 15 min. Immunoassay was developed using TMB Ultra (R&D, Minneapolis, MN, USA) following the manufacturers recommended protocol. Absorbance 450 nm was read using the Biotek Cytation 3 indicating bound his tagged rPro peptide.

### 2.13. Humanization of Murine pro-N-Cadherin Antibody 10A10

Fusion Antibodies, PLC (Belfast, Northern Ireland) was contracted to humanize the murine α-PNC mAb 10A10. CDR regions of the murine α-PNC mAb were cloned onto human IgG_4_ heavy and light chain constant domains. The variable regions of the murine α-PNC mAb were modified in-silico to reduce T-cell epitope antigenicity and increase binding affinity to the prodomain of N-cadherin. These 22 humanized antibody designs were expressed, purified, and ranked for binding affinity using bio-layer interferometry (octet) technology by Fusion Antibodies to the rPro peptide (Appendix A).

### 2.14. Migration Assays

Transwell permeable supports with a 6.5 mm polycarbonate membrane and 8 µm pores were used to separate the upper and lower chambers of a 24-well plate. Both sides of the membrane were coated with 1 µg/mL human plasma fibronectin (Sigma, Saint Louis, MO, USA). Bovine fibronectin was depleted from the complete media using gelatin Sepharose (GE Healthcare, Salt Lake City, UT, USA) following the manufacturers protocol. Complete media containing 10 ng/mL TGF-β1 was then added to the lower chamber at 600 µL/well. Cells were trypsinized, pelleted, resuspended in low serum media containing 0.5% FBS and added into the upper chamber at 1.0 × 10^4^ cells/well in 100μL media per well. After allowing attachment, antibody treatment, rPro or PBS blank was added to the lower chamber and the plate was incubated for 5 h at 37 °C, 5% CO_2_. Media was aspirated, and cells were fixed using 4% Formaldehyde for 15 min at room temperature. Cells were removed from the top of the upper chamber using a sterile cotton swab and wells were washed three times with PBS. Adherent cells on the apical side of wells were then stained by applying DAPI solution (Sigma, Saint Louis, MO, USA) following the manufacturers protocol. Nuclei were visualized by fluorescent microscopy and counted using the Biotek Cytation 3 and latest Gen5 v3.11 software (Winooski, VT, USA). Samples were imaged using the 2.5× objective and total cell numbers were counted in each frame.

### 2.15. Statistical Analysis

Statistical analysis was performed using T-tests and one-way ANOVA with Post hoc multiple comparisons analyses where appropriate using the latest version of GraphPad Prism 9 (San Diego, CA, USA) software (* *p* ≤ 0.05, ** *p* ≤ 0.01, *** *p* ≤ 0.001, **** *p* ≤ 0.0001). Post hoc analyses are indicated in corresponding figure legends. For flow cytometry, chi-squared test was performed using Flowjo v10.7.2 (Vancouver, BC, Canada) analysis software. Chi-squared ≥ 4 is statistically significant.

## 3. Results

### 3.1. pro-N-Cadherin Is Aberrantly Localized in Failing Tissue from Heart, Lung, and Liver and Expressed on the Surface of Isolated Myofibroblasts from Pathological Origins

The role of the classical cadherin CDH2 or N-cadherin is well established in normal cardiac function as well as in the pathogenesis of a number of cardiac diseases and disorders [18,19,20]. Given that N-cadherin signaling is critical for cardiac function and involved in cardiac remodeling, we investigated whether the precursor, proprotein pro-N-cadherin (PNC) plays a role in the pathogenesis of cardiac disease [19]. A group of 15 cardiac tissue samples representing dilated non-ischemic cardiomyopathy, hypertrophic cardiomyopathy, and ischemic infarcted cardiomyopathy were obtained from the Duke University Human Heart Repository (HHR). Samples were deidentified, blinded, and stained for the presence of PNC using a previously characterized murine anti- PNC monoclonal antibody (m-α-PNC mAb) clone 10A10 that binds specifically to the precursor, prodomain of N-cadherin [10]. Each failed, fibrotic tissue shows strong expression of PNC, notably at the intercalated discs where mature N-cadherin would be expected in normal physiology in the heart [21], in the bronchiolar epithelium of fibrotic lungs, and in hepatocytes of fibrotic livers (Figure 1).

To validate the specific presence of aberrantly localized PNC in the case of failing cardiac tissue, a sample of healthy human heart tissue, obtained by surgical excision during a heart transplant procedure, was processed and stained for the presence of PNC, in addition to a human tissue microarray of 24 unique samples of healthy human heart tissue. While cardiac tissue is known to have high protein expression of mature N-cadherin, we observed no aberrantly localized PNC in the healthy hearts. Perinuclear staining found in healthy cardiac tissue is consistent with normal physiology and N-cadherin processing. These data indicate that aberrant PNC localization in the myocardium is a potential biomarker of the failing heart. Tissues that were obtained fresh immediately post-explant were divided, mechanically and enzymatically digested, and isolated as cell monolayers. These isolated fibroblasts were positive for myofibroblast markers α-SMA and type I collagen by flow cytometry (Appendix A). Additionally, all other fibroblast cell lines used expressed α-SMA protein as is expected with the emergence of the myofibroblast phenotype in vitro [22] (Appendix A). Consistent with the aberrant expression observed in the cardiac tissues, we show PNC on the cell surface of myofibroblasts isolated from explant fibrotic cardiac tissue (Figure 2A, DCM-CF), but not on cardiac myofibroblasts isolated from the healthy donor (Figure 2A, NHCF).

To test the generalization of PNC as a potential biomarker for fibrosis, we expanded our investigation to include two additional organs that are commonly associated with fibrosis-related morbidity. A group of lung and liver tissues were examined representing idiopathic pulmonary fibrosis (IPF) and non-alcoholic fatty liver disease with cirrhosis (NAFLD-cirrhosis). Consistent with the phenotype observed in cardiac explant tissue, PNC is aberrantly localized in all fibrotic tissues. All normal tissues examined show no aberrant expression of PNC, indicating that the aberrant localization of PNC is a potential marker of fibrotic pathology (Figure 1). To further validate the cell surface phenotype observed in cardiac myofibroblasts, we obtained two additional commercially available cell lines of IPF etiology, LL97A and LL29, and again show that PNC is present on the surface of myofibroblasts isolated from fibrotic tissues (Figure 2A, DCM-CF, LL97A, LL29) and not on myofibroblasts isolated from healthy tissues (Figure 2A, NHCF, NHLF, CCD-16Lu). Mature N-cadherin was also analyzed in a representative cell line (Appendix A) to confirm no cross reactivity between the PNC antibodies used in these studies (Appendix A) and mature N-cadherin protein. Total lysates of each cell line were immunoblotted for N-cadherin and PNC to corroborate specificity of the PNC antibodies. No cross reactivity between α-PNC antibody and mature N-cadherin is observed (Appendix A). Cell surface proteins were isolated from each cell line and immunoblotted for mature N-cadherin and PNC protein. All cell lines express cell surface N-cadherin as expected; however, PNC is only detected in cell surface protein isolates from myofibroblasts derived from fibrotic tissues (Figure 2B).

Consistent with the flow cytometry results, pathological myofibroblasts immunostained for PNC contain a population of cells with a cell surface PNC positive phenotype. PNC is observed localized to cellular protrusions, indicative of a migratory cell phenotype (Figure 3). The presence and localization of cell surface PNC on cellular protrusions of myofibroblasts isolated from fibrotic tissues suggests PNC is a possible marker of myofibroblasts’ invasive pathology.

### 3.2. pro-N-Cadherin Is Released into Circulation and Quantifiable by Newly Developed ELISA

Mature N-cadherin is solubilized into the plasma under normal physiologic conditions [23]. We hypothesize aberrantly expressed surface PNC is also detectable in the circulation under pathophysiological conditions. We first analyzed the supernatant of cells cultured in vitro for the presence of soluble PNC (sPNC) and feasibility of detection in patient plasma. To verify its presence, sPNC was immunoprecipitated from conditioned media of LL29 myofibroblasts using m-α-PNC mAb 10A10. A 17 kDa product is shown, consistent with the molecular weight of the prodomain of N-cadherin (Figure 4A). We next procured plasma from healthy donors and patients with fibrotic cardiomyopathy, IPF and NAFLD-cirrhosis. The immunoprecipitation of PNC from these samples reveals the 17 kDa prodomain in all plasmas tested and a 60 kDa protein in the plasmas from patients with cardiac fibrosis and NAFLD-cirrhosis (Figure 4A). With these data, we show that PNC is detectable and quantifiable in solution.

After observing solubilized PNC, we sought to optimize and validate a PNC binding sandwich enzyme- linked immunosorbent assay (ELISA). Murine α-PNC monoclonal antibody 10A10 was bound to plates and used as capture antibody and a commercially available biotinylated sheep polyclonal α-PNC antibody was used for the detection antibody. Optimization of capture and detection antibody concentrations was performed using maximal and minimal concentrations of recombinant PNC analyte. A serial dilution of analyte, beginning at 100 ng/mL, was used to determine the range of the standard (Figure 4B). Plasma samples from patients with cardiomyopathy were obtained from the Duke HHR. Dilution linearity of endogenous analyte from three patient plasma samples were analyzed after serial dilution of plasma in standard diluent (Figure 4C). Endogenous analyte from failing heart patient plasma diluted linearly (r^2^ ≥ 0.99). Accuracy of the assay was assessed by recovery of recombinant prodomain-spiked healthy donor plasma relative to the standard back-calculations. Healthy donor plasma diluted 1:1, 1:3, 1:7 and 1:15 in standard diluent was spiked with 10 ng/mL (Figure 4D) and 5 ng/mL (Figure 4E) recombinant prodomain and assayed. All dilutions were quantified and within the accepted range of 20 percent ± the standard back calculations at concentrations 10 ng/mL and 5 ng/mL, validating a novel ELISA for sPNC detection.

### 3.3. Serological Assessment Suggests Cell Surface Release of pro-N-Cadherin from Fibrotic Tissues

To assess sPNC as a potential serological biomarker of fibrosis, we obtained and compared sPNC concentrations in healthy donor plasma to plasma drawn from patients with fibrotic cardiomyopathy. The cohort of samples consists of patients diagnosed with non-ischemic dilated cardiomyopathy, ischemic infarcted cardiomyopathy, and hypertrophic cardiomyopathy with ages ranging from 22 to 74. Plasma from patients with cardiomyopathy contains elevated soluble PNC (sPNC), ranging from approximately 10 ng/mL to 30 ng/mL, while sPNC in healthy donor plasma averages approximately 2 ng/mL (*p* < 0.0001, Figure 4F). Two additional fibrotic conditions representative of lung and liver fibrosis were screened for sPNC. Expression of sPNC in plasma from IPF patients (*n* = 9) and plasma from fibrosis-scored NAFLD-cirrhosis patients (*n* = 12) is significantly elevated relative to healthy donors (*p* < 0.05, Figure 4F). We propose the detection of sPNC could allow for quantitative, non-invasive monitoring of fibrogenesis and disease progression. A large sample prospective study along with clinical analysis will be required for future studies.

### 3.4. Fibronectin Is a Potential PNC Binding Partner

We hypothesize that the prodomain of N-cadherin may bind to an ECM protein component [24]. A common assay to detect binding partners was used to assess binding potential to three major ECM proteins overexpressed in fibrotic conditions. After immobilizing fibronectin, type I and type III collagen to a polystyrene, medium binding plate and blocking with BSA, recombinant his-tagged prodomain (rPro) peptide was incubated over the immobilized and blocked ECM proteins. A biotinylated murine α-his-tag antibody and streptavidin HRP was used for detection of bound rPro peptide. Binding activity of rPro peptide to immobilized fibronectin is shown; however, neither human type I collagen or type III collagen-coated wells efficiently capture rPro from solution (Figure 5A). Subsequently, myofibroblasts isolated from pathological tissues were co-stained for PNC and FN1. Colocalization of PNC and FN1 is shown on cellular protrusions by immunofluorescent microscopy (Figure 5B). Consistent with our binding study, this suggests FN1 is a potential binding partner to the prodomain of N-cadherin.

After observing the binding activity and colocalization of rPro and FN1, we hypothesized that an α-PNC antibody would interfere with this interaction. Fully humanized α-PNC antibody HC5LC4 (Fusion Antibodies, PLC), demonstrates an approximately 10-fold greater affinity to the recombinant prodomain peptide relative to murine 10A10 chimeric control (Appendix A). We therefore selected this clone to test this hypothesis. The FN1 solid phase ELISA protocol (Figure 5A) was used to test m-α-PNC mAb 10A10 and fully humanized α-PNC mAb HC5LC4 (h-α-PNC mAb) blocking activity. Both humanized and murine α-PNC mAbs block rPro peptide from absorbing to immobilized fibronectin (Figure 5C). There is no competition between epitopes for α-His-tag detection antibody and α-PNC antibody (Appendix A). Consistent with the K_d_ values generated by Fusion Antibodies (Appendix A), humanized α-PNC mAb HC5LC4 blocking activity is significantly greater than murine antibody 10A10 (Figure 5C).

### 3.5. Migration of PNC-Expressing Myofibroblasts Is Inhibited by Recombinant Prodomain and Fully Humanized α-PNC Antibody

Given the evidence that PNC is a marker of carcinoma invasion, fibronectin is an inducer of cell motility, and PNC is localized to cellular protrusions on pathological myofibroblasts, we hypothesized that the disruption of the potential cell surface FN1/PNC interaction may prevent the migration of pathogenic PNC expressing myofibroblasts [15,25]. PNC-positive myofibroblasts were subjected to a Boyden chamber migration assay in the presence of h-α-PNC mAb HC5LC4 or human IgG4 isotype control (hIgG4). We also hypothesized treatment with the rPro peptide would impair cell migration by competing with cell surface PNC for binding partners; therefore, treatment with rPro in solution was used as a cell surface PNC antagonist to corroborate the role of cell surface PNC in migration. Boyden chambers were first coated with human FN1 and TGF-β1 was added to the complete media used in the bottom chamber for a chemoattractant. Following a 5 h incubation, migration of PNC-expressing myofibroblasts is significantly impaired by both rPro peptide and h-α-PNC mAb HC5LC4 with no effect on normal myofibroblast cell lines (Figure 6). Interestingly, the reduction in average cell numbers that passed through the membrane is consistent with the percent positivity of cell surface expressing PNC myofibroblasts demonstrated by earlier flow cytometry experiments (Figure 2A). These data suggest that pathological myofibroblasts utilize cell surface PNC to migrate, and targeting cell surface PNC inhibits migration.

### 3.6. Plasma Monitoring of PNC May Be a Useful Clinical Tool

To evaluate the potential for plasma monitoring of PNC to serve as a biomarker for fibrosis development, ELISA-quantified plasma sampled were assessed using AUROC (area under receiver-operating characteristic) analysis and a 95% CI was obtained using the hybrid Wilson/Brown method. Samples were assessed for sensitivity and specificity on a tissue-specific (Figure 7; red, yellow, green lines) and tissue-agnostic (Figure 7; black line) basis. In each case, the AUROC is at least 0.87 (IPF) and in the case of cardiomyopathy, the AUROC is 1.0. While these data are encouraging, we caution that this is a small cohort of retrospective sampling, and further studies will be needed to validate the predictive value of plasma monitoring of PNC in patients with fibrosis.

## 4. Discussion

Our study suggests the need to further investigate N-cadherin biology regarding tissue fibrosis, as reagents developed to study and/or target N-cadherin in this context may or may not cross-react with the pro-N-cadherin precursor protein. This study and others suggest PNC has a unique and divergent role from what is typical of mature N-cadherin [13,15,16,24]. In its mature form, N-cadherin functions as a cell–cell adhesion molecule [10,26]. In its precursor form, the data suggest that it may serve a role in cell-ECM focal adhesions. This is particularly interesting when considering the cardiac structure of the intercalated disc. The major cell adhesion molecule within the fascia adherens of the cardiac intercalated discs is N-cadherin [18,20]. Mature N-cadherin is essential in maintaining the structure and function of the intercalated discs, as well as myofibrillar organization and myocyte shape [19]. Conversely, loss of N-cadherin results in disassembly of intercalated disc structure in the mammalian heart, dilated cardiomyopathy, impaired cardiac function and death [18,27]. The biomechanical consequences of PNC localization to intercalated discs could play a role in the progression of heart failure.

Tissue fibrosis is observed anatomically as the accumulation of excessive extracellular matrix which stresses cell–cell contacts through increased tensile force and elicits maladaptive remodeling of tissues [22,28]. Cells can respond to increased mechanical load at the cell–cell junctions by forming focal adhesions to offload intercellular mechanical force and maintain tensional homeostasis [28,29,30]. Conversely, it is also demonstrated that focal adhesions and cell–cell junctions can function inversely [31]. In the heart, this is observed in developmental and pathological processes. Mimicking fibrosis in cardiac cells resulted in increased focal adhesion formation adjacent to the cell–cell interface [28]. This was interpreted as the need for mechanical off-loading from the intercalated discs. In the pathological processes of cardiac fibrosis and remodeling, increased focal adhesion is correlated with lower working efficiency of the cardiac tissue [28]. The observation of FN1 as a potential PNC binding partner leads us to postulate that post translational changes to N-cadherin processing, through retention of the prodomain could serve as a method for poised cellular response to changes in mechanical load. Future studies are required to understand the role of PNC at the cell surface in settings of fibrosis.

It is possible the role of PNC is cell dependent. In this study, PNC expression is observed in multiple different diseased tissues associated with fibrosis and tissue remodeling. Targeting PNC with a monoclonal antibody specific for an epitope on the prodomain results in reduction of migration by PNC expressing myofibroblasts suggesting a role for PNC in myofibroblast migration. This is consistent with the role of cell surface PNC in carcinogenesis and a recent report that cell surface PNC regulates apico-basal polarity in neural stem cells [13,15,16,24]. Cell surface PNC expression in vivo, but not mature N-cadherin, caused loss of apico-basal polarity, breach of basal membrane and invasion of neural precursors into the ventricle and surrounding mesenchyme of the neural tube [13]. The observation that rPro antagonizes cell migration suggests cell surface PNC is a mediator of pathological myofibroblast migration, as opposed to the sPNC. In progressive fibrosis, myofibroblasts continually migrate into and within the affected organ without resolving, leading to disease progression. This is readily apparent in cardiac fibrosis and IPF disease progression [22,32]. In the case of cardiac fibrosis, myofibroblasts can be found widespread within the failing heart [22]. It is thought that IPF is initiated at the periphery of the lungs and slowly migrates to encompass the entirety of the lungs [33]. It has been demonstrated that myofibroblasts isolated from advanced IPF display a higher capacity to migrate than those isolated from less advanced disease stages [34]. Targeting myofibroblasts migration is a reasonable therapeutic strategy for mitigating fibrotic conditions.

It has been postulated that cell surface prodomain cleavage from N-cadherin is a means of spatially and temporally regulating adhesion during development and synapse formation [14]. Increased expression and proteolysis of the prodomain at the cell surface could explain elevated levels of sPNC found in the plasma of patients with pathological fibrosis. It could also be argued that increased cell death due to disease could contribute to the elevation of the 17 kDa prodomain in the plasma. Almost one third of the plasma proteome is made up of intercellular proteins that have escaped due to cellular turnover [35]. With the origins of the 60 kDa protein unknown, future studies are necessary to understand the processing of PNC in tissue fibrosis.

Our study is a preliminary report with a relatively limited and mostly retrospective sample set, highlighting the need for further studies to thoroughly evaluate PNC as both a biomarker and a therapeutic target in fibrotic disease. Additionally, our data (Figure 1) indicate that epithelial cells in each respective tissue and not only the myofibroblasts may have an important role in PNC-driven fibrotic disease. This is important considering existing work that identifies PNC as a driver of migration and invasion in carcinogenesis [15,16].

## 5. Conclusions

In this work, we identified PNC as a potential biomarker of fibrotic conditions, PNC was shown to be detectable in the plasma of patients with cardiomyopathy, IPF and NAFLD-cirrhosis in a quantifiable manner and we have suggested that aberrant proteolytic processing of N-cadherin is involved in the pathophysiology of fibrosis. Perhaps most importantly, PNC is expressed on the cell surface of myofibroblasts and potentially other cells within tissues in the pathological setting of fibrosis, but has not been observed on the surface of physiologically normal postnatal cell types [10,14,15]. This makes PNC an attractive candidate for a potential serological surrogate for fibrosis progression and potential therapeutic target that may offer specificity to failing tissues and the invasive myofibroblasts that drive disease. Further studies will be required to discern to what extent serological concentration of sPNC correlates with disease progression. In summary, pro-N-cadherin is aberrantly localized in all fibrotic tissues examined in this study and elevated in the plasma of all patients with fibrosis analyzed in this study.

## 6. Patents

Authors PDF and SVP have been awarded US patent 11,022,608, 2021.

## Figures and Tables

**Figure 1 cells-11-00156-f001:**
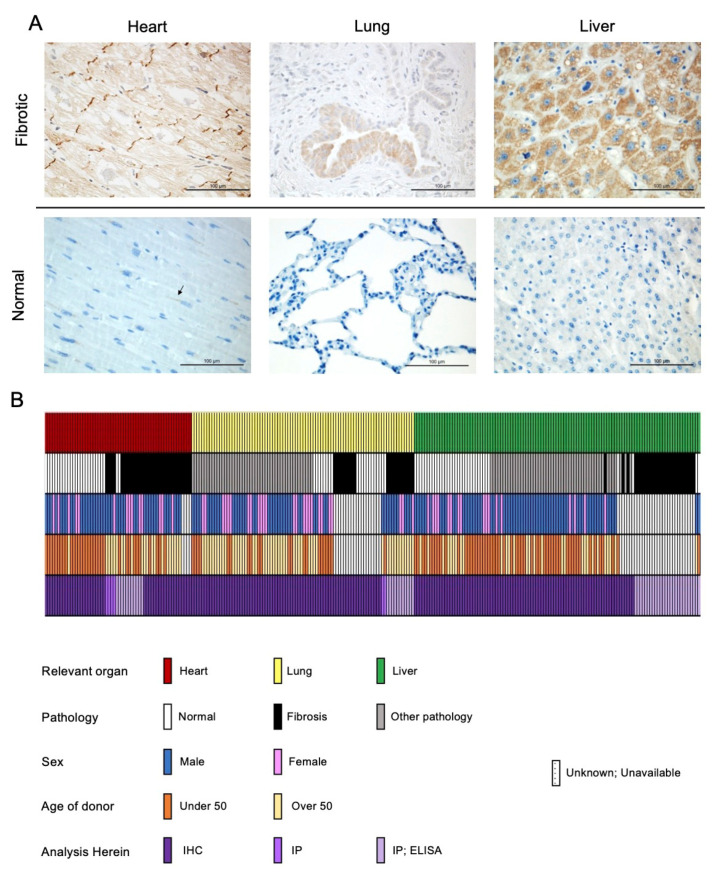
PNC is aberrantly expressed in fibrotic heart, lung, and liver tissues. (**A**) Representative images of stained tissues from explanted failed human hearts (*n* = 20, ischemic and non-ischemic etiology), lungs (*n* = 10, IPF etiology) and livers (*n* = 40, NAFLD-cirrhosis etiology) show positive expression and aberrant localization of PNC (Brown stain). Corresponding representative images of stained normal human heart (*n* = 24), lung (*n* = 18), and liver (*n* = 32) show lack of aberrantly expressed PNC at the tissue level. Perinuclear staining, consistent with normal N-cadherin processing can be seen in the healthy cardiac tissue. Scale bar = 100 µm. (**B**) Graphical representation of human samples analyzed in this study. Each column represents a single human sample, annotated by color to indicate the organ system of relevance to this study (top row), whether the patient had fibrosis (second row), the reported sex of the patient (third row), the age of the patient (fourth row) and the method by which the sample was analyzed in this study (fourth row). In cases where demographic data is unknown, unavailable, or unreported, a white bar with black hatch is shown. *n* = 257 total samples analyzed.

**Figure 2 cells-11-00156-f002:**
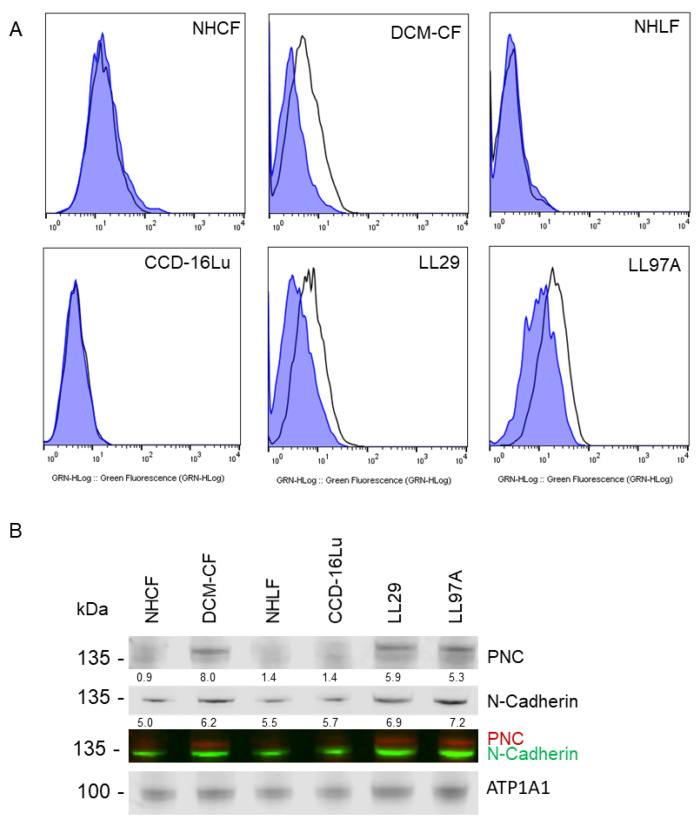
PNC is localized to the cell surface of myofibroblasts. Myofibroblasts from heart and lung tissues were stained and analyzed by flow cytometry using m-α-PNC mAb 10A10, excluding debris and dead cells via gating and 7AAD exclusion using Flowjo software. Myofibroblasts from fibrosis origins stain positive for cell surface PNC: (**A**) CF-DCM, cardiac myofibroblasts from dilated cardiomyopathy; Chi-Squared = 177.5; SE Dymax % Positive = 48.0. LL97A, IPF; Chi-Squared = 93.4; SE Dymax % Positive = 54.8, and LL29, IPF; Chi-Squared = 113.5; SE Dymax % Positive = 51.0. PNC was not detected on the surface of primary normal human cardiac myofibroblasts from healthy donor, NHCF; Chi-Squared = 3.47; SE Dymax % Positive = 8.54, primary normal human lung myofibroblasts from healthy donor, NHLF; Chi-Squared = 0; SE Dymax % Positive = 0, or immortalized CCD-16Lu lung myofibroblasts from healthy donor; Chi-Squared = 0; SE Dymax % Positive = 0. Mouse IgG1 isotype control (blue shaded) was compared to m-α-PNC mAb (unshaded). Results are representative of 3 independent experiments, *n* = 3. For all flow cytometry experiments, Chi-squared ≥ 4 is statistically significant. (**B**) Cell surface proteins were isolated from myofibroblasts and immunoblotted for N-cadherin, PNC, and Na,k-ATPase α-1 (ATP1A1) cell surface compartment loading control. PNC and N-Cadherin lysates were normalized to the ATP1A1 cell surface loading control for each sample and are reported as a relative value below each band.

**Figure 3 cells-11-00156-f003:**
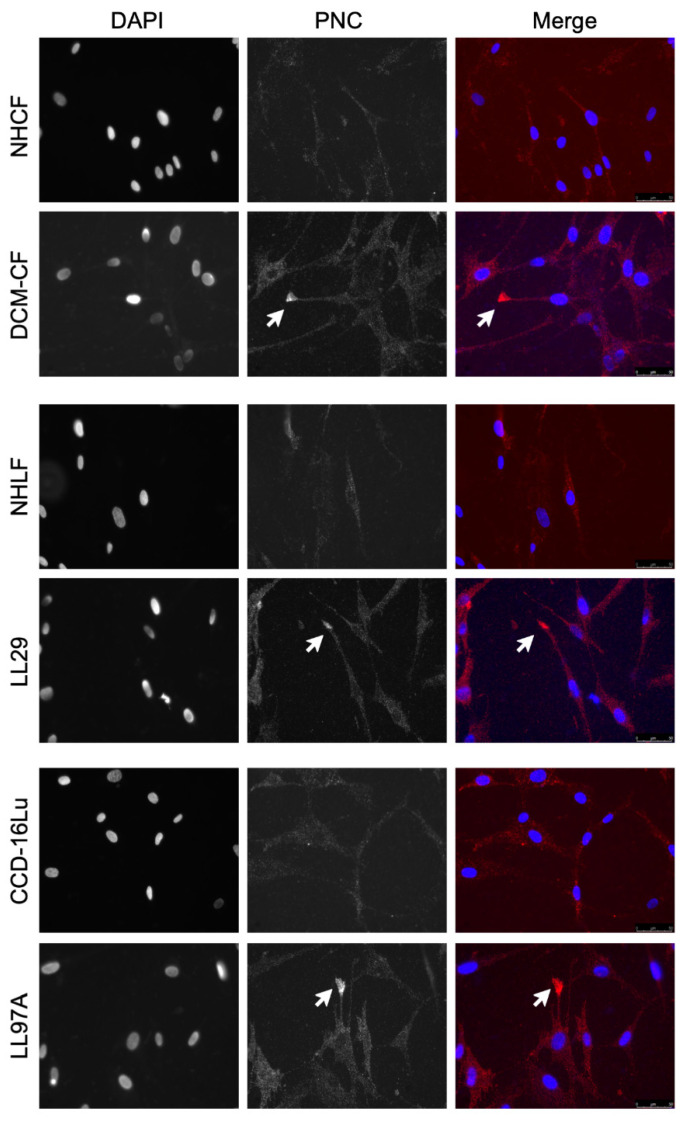
PNC is localized to cellular protrusions. Fixed, unpermeabilized myofibroblasts from fibrotic and healthy tissues were immunostained for m-α-PNC mAb 10A10 (red) and DAPI (blue). PNC is localized to cellular protrusions on pathological myofibroblasts DCM-CF, LL29, and LL97A (arrows). PNC is not expressed on the surface of NHCF, NHLF, or CCD-16Lu isolated from healthy donor.

**Figure 4 cells-11-00156-f004:**
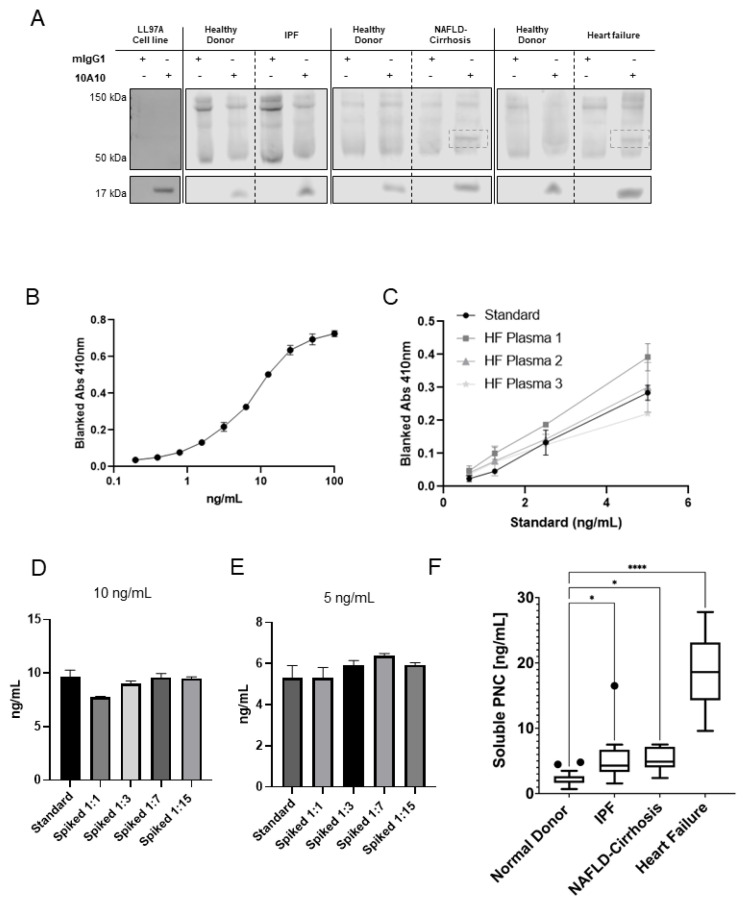
Feasibility and development of a PNC ELISA. (**A**) Soluble PNC product was immunoprecipitated from LL29 conditioned media and patient plasma from pooled healthy donors, IPF patients, NAFLD-Cirrhosis patients and cardiomyopathy patients using m-α-PNC mAb 10A10 compared to mouse IgG1 isotype control (mIgG1) and immunoblotted. (**B**) Recombinant prodomain analyte was serially diluted in duplicate starting at 100 ng/mL in standard diluent and measured by ELISA to determine the range of the standard. (**C**) Linearity of endogenous analyte was measured. Plasma from patients with heart failure was serially diluted 1:3, 1:7, 1:15, 1:31 in standard diluent and sPNC was measured by ELISA in duplicate. All linear regression r^2^ values were calculated to be within acceptable range (r^2^ ≥ 0.99). (**D**,**E**) Healthy donor plasma was diluted in standard diluent to indicated dilutions (1:1, 1:3, 1:7, 1:15) then spiked with recombinant prodomain analyte at 10 ng/mL (**D**) and 5 ng/mL (**E**) and analyzed by ELISA. Recovery of analyte was quantified and compared to back calculation of the standard. All dilutions were within the consensus range of 20 percent ± the standard back calculations at concentrations 10 ng/mL and 5 ng/mL. (**F**) Plasma was assayed for sPNC from healthy controls (*n* = 26), patients with IPF (*n* = 9), NAFLD with cirrhosis (*n* = 12), and cardiomyopathy (*n* = 9). Ordinary one-way ANOVA analysis with Dunnett’s multiple comparisons test was performed to determine significance (* *p* ≤ 0.05, **** *p* ≤ 0.0001).

**Figure 5 cells-11-00156-f005:**
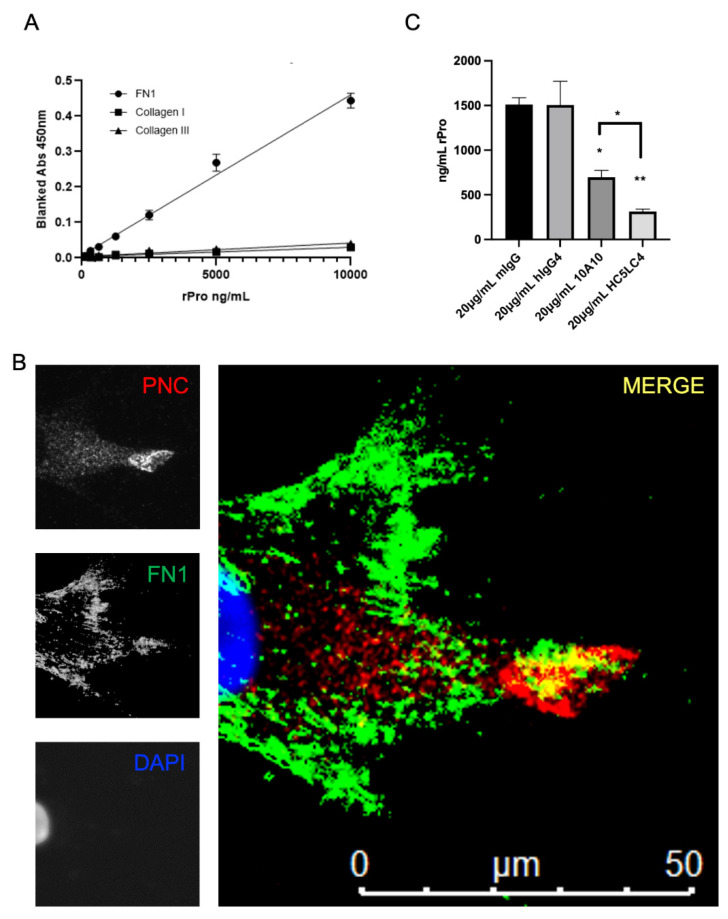
FN1 is a potential PNC binding partner. (**A**) Medium binding ELISA plates were coated with human fibronectin, collagen type I, or collagen type III, then blocked and incubated with his-tagged recombinant N-cadherin prodomain. After washing unbound prodomain, prodomain binding to the immobilized substrate was measured using a biotinylated his-tag specific monoclonal antibody and streptavidin-HRP for detection. Assay was performed in quadruplicate and is representative of at least 3 independent experiments. (**B**) Representative image of a cardiac myofibroblast isolated from failed cardiac explant tissue showing colocalization of PNC and FN1 immunostained for PNC (red), FN1 (green) and DAPI (blue). Yellow indicates PNC/FN1 colocalization (Merge). (**C**) Medium binding ELISA plates were coated with fibronectin, blocked, then incubated with either recombinant prodomain of N-cadherin or prodomain in combination with mouse IgG1 isotype control, human IgG4 isotype control, m-α-PNC mAb 10A10 or h-α-PNC mAb HC5LC4. Bound recombinant prodomain was detected using anti-his-tag monoclonal antibody in technical duplicates and representative of at least 3 independent experiments (*n* = 3). Ordinary one-way ANOVA analysis with Tukey’s multiple comparisons test was performed to determine significance (* *p* ≤ 0.05, ** *p* ≤ 0.01).

**Figure 6 cells-11-00156-f006:**
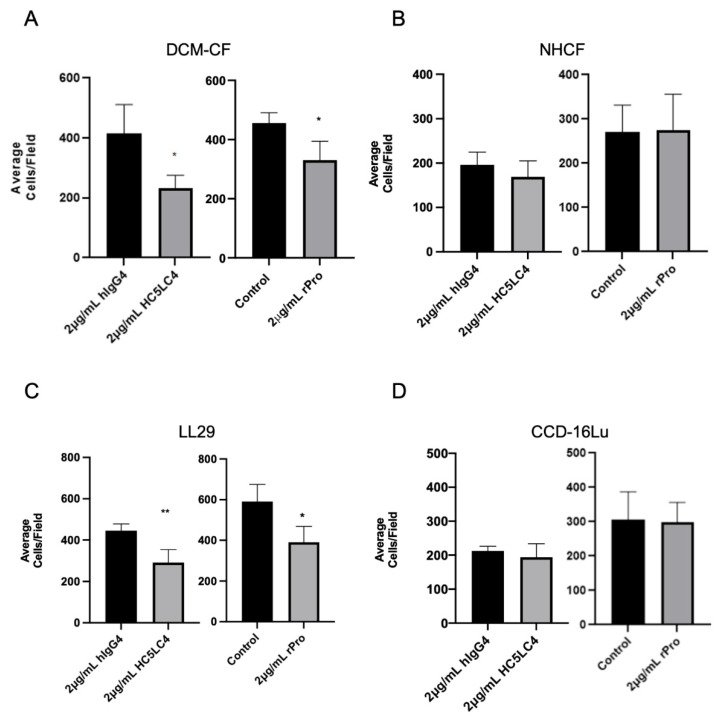
Cell surface PNC has a role in myofibroblast migration. Transwell permeable supports with a 6.5 mm polycarbonate membrane and 8 µm pores were coated with human fibronectin and used to separate the upper and lower chambers of a 24-well cell culture plate to measure migration of cells across the membrane (*n* = 4). Pathological myofibroblast migration is significantly reduced by h-α-PNC mAb (HC5LC4) and recombinant prodomain of N-cadherin (rPro) after 5 h; (**A**) DCM-CF, dilated cardiomyopathy myofibroblasts (**C**) LL29, IPF myofibroblasts. No significant effect on myofibroblasts isolated from healthy tissues was observed; (**D**) immortalized CCD-16Lu lung myofibroblasts from healthy donor (**B**) NHCF, normal human cardiac myofibroblasts. Two tailed T-test assuming Gaussian distribution analysis was performed to determine significance. (* *p* ≤ 0.05, ** *p* ≤ 0.01).

**Figure 7 cells-11-00156-f007:**
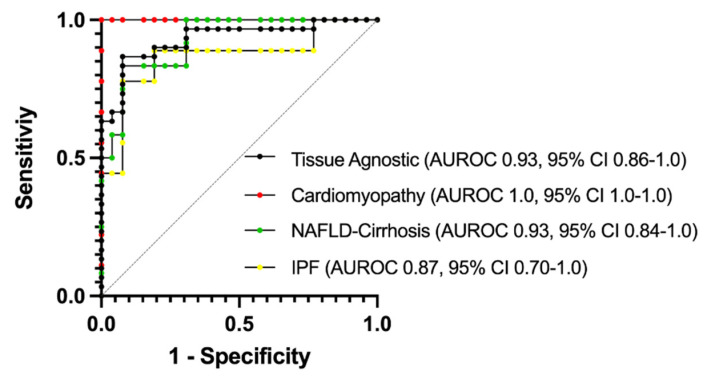
AUROC analysis of plasma PNC for specificity and sensitivity. Normal donor samples (*n*= 26) were compared to samples from cardiomyopathy (Red, *n* = 9), NAFLD-Cirrhosis (Green, *n* = 12) and IPF (Yellow, *n* = 9) patients. To determine tissue-agnostic AUROC, all fibrotic samples were combined (Black, *n*= 30) and compared to normal donor samples. AUROC = area under the receiver operating characteristics curve.

## Data Availability

The data presented in this study are contained within the article and Appendix A. Murine mAb 10A10 is openly accessible via ATCC Patent depository (atcc.org (accessed on 6 December 2021)).

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
