# Peer review of "Pathologic Proteolytic Processing of N-Cadherin as a Marker of Human Fibrotic Disease"

_cells, 2022, doi:10.3390/cells11010156_

Round 1
Reviewer 1 Report
This manuscript describes the proprotein pro-N-Cadherin (PNC) as a marker of human fibrotic disease of heart, lung and liver and it suggested that aberrant PNC is involved in the pathophysiologiy of fibrosis. Furthermore the authors suggested that PNC has a unique and divergent role from what is typical of mature N-cadherin, probably contributing in cell-ECM focal adhesions. The manuscript is well-written with a clear message and the findings are very interesting. However there are some minor advices that need to be performed:
- Please normalize the immunoblotted results in Figure 2B
- I suggest to show the result with RPL13A
Reviewer 2 Report
The investigation shows evidence that pro-N-cadherin could be playing a key role in the progression of different fibrotic diseases including that developed in the heart, lung, and liver of humans. The finding is of relevance since also strongly suggests that detection of abnormal levels of pro-N-cadherin in plasma of patients bearing fibrotic diseases could be a diagnosis biomarker of the abnormal fibrogenesis. Nonetheless, this version of the manuscript still needs some minor improvements before proceeding further.
To propose the proteolytic processing of N-Cadherin as a plasma biomarker of human fibrotic diseases, it is highly suggested that authors include sensitivity and specificity data such as that obtained through ROC curve analyses. These analyses will give greater credibility and impact to the proposal.
The authors should discuss about the reduced number of samples as an important limitation of the investigation. In addition, the clinical relevance of the finding should also be highlighted.
A missing word in the abstract section: Finally, we have humanized a murine antibody and demonstrate THAT it significantly inhibits migration of pro-N-cadherin expressing myofibroblasts.
The catalog number of some antibodies are missing such as that of α-SMA antibody (Cell Signaling), at Cell Culture and Reagents section
Kilodalton symbol should be kDa instead of kD. Correction should be made both in the text and figures.
The 60 kDa band cited at page 13, lines 411 should be pointed out in the images of figure 4A. This indication will make it easier for readers.
Reviewer 3 Report
The authors hypothesized abnormal processing of N-cadherin as a pathogenic mechanism in fibrosis and showed that the precursor form of N-cadherin (pro-N-cadherin; PNC) is aberrantly expressed in fibrotic tissues. Further, this form was detected in the plasma of the patients with fibrotic heart, liver and lung diseases. Together, they claim the usefulness of pro-N-cadherin/prodomain of N-cadherin as a potential biomarker for these fibrotic diseases. Overall, the authors showed convincing evidence to support their conclusion, but several concerns remain to be resolved.
Comments:
- The authors described IHC analysis of PNC in the failing heart, but no detailed description in the fibrotic lung and liver (Fig. 1A). In the lung and liver bronchiolar epithelial cells and hepatocytes appear to be strongly stained, respectively, suggesting that these two cell types are major targets. Given that, it is recommended that the authors show PNC staining in the fibrotic foci of the lung and liver.
Furthermore, the authors used cultured fibroblasts as a model system for investigation of PNC (Fig. 2 & 3). What would be the potential role of PNC in epithelial cells? Does it disrupt cell-cell adhesion? Does it induce EMT and cell migration?
- In Fig. 4A, the authors mentioned 64 kD protein that is specifically recognized in the patients with fibrotic diseases. However, this band does not appear in IPF. I just wonder whether this 64 kD protein is derived from PNC. Any further evidence?
- In Fig. 4F, Soluble plasma PNC values from IPF and NAFLD-cirrhosis appear to be marginally higher than those from normal donors. I would not say that these data strongly support a generalization of soluble PNC as a marker for three types of fibrotic diseases.
- In Fig. 6, there must be a correction in the legend as follows. (A), DCM-CF; (B), NHCF; (C), LL29; (D), CCD-16Lu.
